# Machine Learning Approach to Select Small Compounds in Plasma as Predictors of Alzheimer’s Disease

**DOI:** 10.3390/ijms26146991

**Published:** 2025-07-21

**Authors:** Eleonora Stefanini, Alberto Iglesias, Joan Serrano-Marín, Juan Sánchez-Navés, Hanan A. Alkozi, Mercè Pallàs, Christian Griñán-Ferré, David Bernal-Casas, Rafael Franco

**Affiliations:** 1Molecular Neurobiology Laboratory, Department of Biochemistry and Molecular Biomedicine, Universitat de Barcelona, 08028 Barcelona, Spain; eleonora.stefanini.2028@student.uu.se (E.S.); alberto.iglesias@alumni.urv.cat (A.I.); joan.serrano.marin@gmail.com (J.S.-M.); 2Department of Ophthalmology, Oftalmedic, I.P.O. Institute of Ophthalmology, 07011 Palma de Mallorca, Spain; juansanchez.naves@gmail.com; 3Department of Optometry, College of Applied Medical Sciences, Qassim University, Buraydah 51452, Saudi Arabia; h.alkozi@qu.edu.sa; 4Departament de Farmacologia i Química Terapeùtica, Universitat de Barcelona, 08028 Barcelona, Spain; pallas@ub.edu (M.P.); christian.grinan@ub.edu (C.G.-F.); 5Institut de Neurociències, Universitat de Barcelona, 08035 Barcelona, Spain; 6CiberNed, Network Center for Neurodegenerative Diseases, Spanish National Health Institute Carlos III (ISCIII), 28029 Madrid, Spain; 7Department of Information and Communication Technologies, Universitat Pompeu Fabra, 08018 Barcelona, Spain; bernalcdub1@gmail.com; 8School of Chemistry, Universitat de Barcelona, 08028 Barcelona, Spain

**Keywords:** metabolomics, biomarker, neurodegenerative disease, blood, diagnosis

## Abstract

This study employs a machine learning approach to identify a small-molecule-based signature capable of predicting Alzheimer’s disease (AD). Utilizing metabolomics data from the plasma of a well-characterized cohort of 94 AD patients and 62 healthy controls; metabolite levels were assessed using the *Biocrates MxP^®^ Quant 500* platform. Data preprocessing involved removing low-quality samples, selecting relevant biochemical groups, and normalizing metabolite data based on demographic variables such as age, sex, and fasting time. Linear regression models were used to identify concomitant parameters that consisted of the data for a given metabolite within each of the biochemical families that were considered. Detection of these “concomitant” metabolites facilitates normalization and allows sample comparison. Residual analysis revealed distinct metabolite profiles between AD patients and controls across groups, such as amino acid-related compounds, bile acids, biogenic amines, indoles, carboxylic acids, and fatty acids. Correlation heatmaps illustrated significant interdependencies, highlighting specific molecules like carnosine, 5-aminovaleric acid (5-AVA), cholic acid (CA), and indoxyl sulfate (Ind-SO_4_) as promising indicators. Linear Discriminant Analysis (LDA), validated using Leave-One-Out Cross-Validation, demonstrated that combinations of four or five molecules could classify AD with accuracy exceeding 75%, sensitivity up to 80%, and specificity around 79%. Notably, optimal combinations integrated metabolites with both a tendency to increase and a tendency to decrease in AD. A multivariate strategy consistently identified included 5-AVA, carnosine, CA, and hypoxanthine as having predictive potential. Overall, this study supports the utility of combining data of plasma small molecules as predictors for AD, offering a novel diagnostic tool and paving the way for advancements in personalized medicine.

## 1. Introduction

Alzheimer’s disease (AD) is a progressive neurodegenerative disorder and the most common cause of dementia worldwide, affecting over 55 million individuals and posing a growing public health crisis as populations age [1]. Pathologically, AD is characterized by the accumulation of amyloid-beta (Aβ) plaques, neurofibrillary tangles composed of hyperphosphorylated tau protein, synaptic dysfunction, and chronic neuroinflammation [2,3].

Despite decades of research, the precise mechanisms driving AD pathogenesis remain elusive, and current therapeutic strategies, primarily targeting Aβ aggregates, NMDA receptors, and acetylcholinesterase, have shown limited clinical efficacy and/or adverse effects [2]. This underscores the urgent need for a deeper understanding of the molecular underpinnings of AD and the development of robust diagnostic tools to enable early intervention.

Metabolomics has emerged as a powerful approach to unravel the complex metabolic perturbations associated with AD, offering insights into dysregulated pathways beyond classical amyloid and tau pathology [4]. Metabolites, as downstream products of cellular processes, reflect the functional state of organs and tissues, including the brain. A recent study, have leveraged targeted metabolomics to systematically evaluate metabolic changes in AD across plasma and brain tissue [5]. Using the *Biocrates MxP^®^ Quant 500* platform, the authors analyzed 630 metabolites (106 small molecules and 524 lipids) in two cohorts: 158 plasma samples (94 AD patients, 64 controls) and 71 postmortem frontal cortex samples (35 AD, 36 controls). Their results have revealed widespread metabolic disruptions in AD, implicating oxidative stress, mitochondrial dysfunction, gut–brain axis alterations, and lipid metabolism dysregulation. In addition, shifts in plasma levels of bile acids, indole derivatives, and polyunsaturated fatty acids are identified in AD patients, suggesting potential roles for gut microbiota and systemic inflammation in disease progression [5].

While such studies advanced our understanding of AD-associated metabolic disruptions, translating these findings into clinically actionable tools requires a focused effort to optimize diagnostic accuracy and reproducibility. The primary aim of our study diverges from prior exploratory metabolomic analyses by specifically targeting diagnostic categorization using plasma samples. Building on the dataset of Kalecký et al. (2022) [5], we applied advanced normalization techniques and machine learning-based feature selection to identify optimal metabolite combinations capable of distinguishing AD patients from healthy controls with high precision. Our approach leverages recent methodological innovations by Serrano-Marín et al., which enhance the reliability of metabolomic data analysis and enable the development of multivariate diagnostic models [6]. By employing linear discriminant analysis (LDA) and rigorous cross-validation, we identified metabolite panels that achieve diagnostic accuracy exceeding 75%, emphasizing the integration of metabolites showing opposing trends in AD (e.g., increased carnosine and decreased cholic acid).

This diagnostic framework advances the practical application of metabolomics in AD and paves the way for personalized medicine strategies, where treatments could be tailored based on individual metabolic profiles. Furthermore, our findings reinforce the importance of systemic metabolic dysregulation in AD pathogenesis, offering new avenues for therapeutic targeting beyond traditional amyloid-centric approaches.

## 2. Results

### 2.1. Correlations Between Data from Patients and Controls

#### 2.1.1. Dataset Overview and Data Cleaning Outcome

The initial dataset consisted of 158 plasma samples (94 AD patients and 64 control subjects) with both lipid and small molecule metabolite data, which was retrieved from supplementary information of a previous study [5]. After removing two low-quality samples (PX009 and PX025) for reasons specified in our previous study [7], the dataset was reduced to 156 samples (94 AD patients and 62 controls). Initially, lipid datasets were also considered for the analysis. Apart from the findings of elevated levels of CE(20:4) and CE(22:5) cholesteryl ester in the plasma of AD patients with ε3/ε4 genotype, the lipid datasets did not led to any promising result in terms of group categorization [7]. We then considered the remaining eight biochemical families. As indicated in Methods, preprocessing led to removal of samples for which multiple missing values were reported due to being “below the limit of detection (LOD)”. The final datasets contained the data corresponding to eight biochemical groups: amino acids (150 samples, 20 metabolites), amino acid–related (154 samples, 30 metabolites), bile acids (152 samples, 14 metabolites), biogenic amines (152 samples, 14 metabolites), indoles and derivatives (155 samples, 4 metabolites), carboxylic acids (150 samples, 8 metabolites), hormones and related compounds (154 samples, 4 metabolites), and fatty acids (154 samples, 12 metabolites). Although all eight groups were initially considered, data for amino acids and hormones did not lead to sufficient predictive performance in downstream analyses and were then excluded. Data corresponding to the remaining six chemical families are provided in Appendix A.

#### 2.1.2. Selection of the *Metabolite_c* for Each Biochemical Family

A concomitant metabolite (*metabolite_c*) was selected within each biochemical group using the procedure described in the Section 4. The selection was based on a balance of statistical performance and biological relevance, with particular focus on the adjusted *R*^2^ values and F-test *p*-values obtained from the linear regression models. In biological studies, an adjusted *R*^2^ value between 0.2 and 0.4 is considered a fair to good indication of model fit and is commonly encountered when analyzing complex biological data. While some models did not achieve statistically significant *p* or optimal *R*^2^ values, the selection process carefully integrated both statistical performance and biological plausibility. This approach recognizes that perfect statistical significance is not always attainable in biological contexts, but the overall model performance and biological interpretability were critical in selecting the most relevant concomitant metabolite.

Using the procedure described in Methods and taking *R*^2^ values for reference, the following metabolites *metabolite_c* were selected for using its value as concomitant parameter: Symmetric dimethylarginine (SDMA) for the amino acid-related group, Taurodeoxycholic acid (TDCA) for bile acids, Spermine for biogenic amines, 3-Indoleacetic acid (3-IAA) for indoles, Succinic Acid (Suc) for carboxylic acids, and Docosahexaenoic acid (DHA) for fatty acids (see Table 1).

#### 2.1.3. Residual Correlation Heatmaps

For each of the six biochemical families in which a concomitant metabolite (*metabolite_c)* was selected, correlation heatmaps of the residuals were generated to explore inter-metabolite relationships beyond known covariates. Residuals were calculated as the difference between the observed and predicted log_10_-transformed metabolite values. Predicted values obtained from the previously described linear regression model. Pearson correlation coefficients were computed on these residuals, and the resulting heatmaps visualize the pairwise correlations. In each heatmap, the color of the metabolite label reflects the direction of change between AD patients and controls: red indicates a tendency to increase in AD, blue a tendency to decrease in AD. Dashed boxes highlight metabolites whose group-associated *p*-values (from the regression model) are <0.05, while solid boxes indicate significance after multiple testing correction (*p*-adj < 0.05). These highlighted metabolites are those that, individually, showed a statistically significant association with disease status and may theoretically have greater discriminative potential between patients and controls.

The main findings observed within each biochemical family are described below.

#### 2.1.4. Amino Acid Related

In the amino acid-related family, the residual correlation heatmap (Figure 1) revealed generally weak pairwise associations, i.e., no strong positive or negative correlations. However, a subtle predominance of positive over negative correlations was observed. Residuals for 3-Methylhistidine (3-Met-His) and 1-Methylhistidine (1-Met-His) displayed a modest positive correlation, which was also evident for taurine and carnosine and for cystine and DOPA. Interestingly, ornithine showed a positive association with citrulline, alpha-aminoadipic acid (alpha-AAA), asymmetric dimethylarginine (ADMA), and sarcosine, compounds that are connected via the urea cycle and methylation pathways. The only notably negative correlation observed was between methionine-sulfoxide (Met-SO) and cystine.

Regarding the metabolite-specific differences between the clinical groups, both “up”- and “down”-regulated metabolites in AD were evenly distributed in this biochemical group, as indicated by the high number of metabolite names appearing as red and blue in Figure 1. The metabolites for which correlations reached significance before adjustment (*p* < 0.05; within the dashed frame in Figure 1), were taurine, trans-4-hydroxyproline (t4-OH-Pro), and 1-methylhistidine (1-Met-His) that tend to increase in AD and cystine and betaine that tend to decrease. Following multiple testing correction (*p*-adj < 0.05; within the solid frame in Figure 1), correlations were significant for carnosine and 5-aminovaleric acid (5-AVA), both with tendency to increase in AD, and for DOPA, which tends to decrease in AD.

#### 2.1.5. Bile Acids

The bile acid correlation heatmap (Figure 2) showed patterns of co-variation, with some metabolites forming closely related groups. Patterns of co-variation in disease were detected for conjugated bile acids: glycocholic acid (GCA), glycochenodeoxycholic acid (GCDCA), taurocholic acid (TCA), and taurochenodeoxycholic acid (TCDCA), which showed strong positive correlations. Cholic acid (CA) and chenodeoxycholic acid (CDCA) exhibited group-specific correlation patterns (*p* < 0.05; within the dashed frame in Figure 2), with CA displaying significant differences after multiple testing correction (*p*-adj < 0.05; within the solid frame in Figure 2).

#### 2.1.6. Biogenic Amines

The heatmap of biogenic amines (Figure 3) revealed both positive and negative correlations. A positive correlation emerged between putrescine and spermidine and between choline, trimethylamine N-oxide (TMAO), and hypoxanthine. In contrast, a negative correlation was identified between spermidine and histamine. Interestingly, hypoxanthine, a metabolite of purine catabolism and precursor of uric acid, tended to decrease in AD. The other metabolites in the family with a tendency to decrease in AD were trigonelline and choline. TMAO and hypoxanthine were the only ones within the dashed frames (Appendix A), indicating that the residuals of metabolites exhibit significant differences between patients and controls based on their raw *p*-values (*p* < 0.05). Significance was not reached after multiple testing correction.

#### 2.1.7. Carboxylic Acids

In this group of metabolites, all but hippuric acid (HipAcid) tended to increase in AD. 3,4-Dihydroxyphenylacetic acid (OH−GlutAcid) demonstrates a modest positive correlation with aconitic acid (AconAcid), while exhibiting a weak negative correlation with all other metabolites. Notably, OH−GlutAcid is the only metabolite enclosed within a significance grid (*p* < 0.05; dashed frame in Figure 3). However, no significance remained after multiple testing correction.

#### 2.1.8. Indoles

In the indole biochemical group, correlation analysis of residuals revealed weak associations among metabolites. A slight negative correlation was observed between indole and 3-indolepropionic acid (3-IPA), while indole and indoxyl sulfate (Ind-SO_4_) exhibited a modest positive correlation, suggesting a certain degree of co-regulation. Indole and Ind-SO_4_ levels tend to increase in AD, while 3-IPA levels tend to decrease. Of the three indoles analyzed, both 3-IPA and Ind-SO_4_ reached nominal significance (*p* < 0.05), but only Ind-SO_4_ remained significant after multiple testing correction (*p*-adj < 0.05; solid frame in Figure 4). Despite the generally weak correlations observed, the opposing directionality of these metabolites may indicate a shift from protective to potentially harmful indole derivatives in AD.

#### 2.1.9. Fatty Acids

A prominent cluster of positive correlations is evident among the fatty acids, particularly between octadecenoic acid (FA(18:1)) and octadecadienoic acid (FA(18:2)), indicating coordinated changes despite opposite trends in abundance between the AD and control groups. Of particular interest is eicosapentaenoic acid (EPA), being the only fatty acid in the group whose residuals show no positive correlation with those of the other metabolites. Arachidonic acid (AA) tends to increase in AD and its residuals positively correlate with those of nearly all other fatty acids in the group. Arachidonic acid (AA) is the only metabolite showing significance before and after multiple testing correction (*p*-adj < 0.05; solid frame in Appendix A).

### 2.2. Classification Performance via LDA

#### 2.2.1. Univariate Analysis

Univariate linear discriminant analysis (LDA) models, constructed separately for residuals of each metabolite, demonstrated discriminative performance with classification accuracies ranging from 0.59 to 0.63. Aiming at obtaining accuracies higher than 0.75, multivariate models that combined residuals of two to five metabolites were checked. The system was trained to increase the accuracy while optimizing both sensitivity and specificity.

#### 2.2.2. Multivariate Analysis: Four Metabolite Combinations

Table 2 presents combinations of four metabolites that achieve an accuracy of ≥0.75, along with their respective metabolic families and their direction of change in AD, increase (in red) or decrease (in blue). Notably, carnosine appears in all but one of these combinations. Similarly, 5-AVA is included in several combinations. Strikingly, no fatty acids are featured in any of the high-accuracy sets. Only two combinations include members of the carboxylic acid family (lactic and hippuric acids).

Importantly, every combination includes metabolites that increase in AD as well and those that tend to decrease, underscoring the relevance of integrating opposing trends. The combination with the highest sensitivity score (0.80) includes metabolites from four different families: 5-AVA (amino acid-related), CA (bile acids), serotonin (biogenic amines), and Ind-SO_4_ (indoles).

We evaluated whether the inclusion of an additional metabolite—selected based on its statistical significance in the heatmaps (*p*-adj < 0.05)—could enhance the performance of the multivariate models. Specifically, we added indoxyl sulfate (Ind-SO_4_) to the best-performing four-metabolite combination (5-AVA, carnosine, CA, and CDCA). However, this five-metabolite model did not yield improved classification performance and, in some cases, showed even poorer results compared to the original four-component model.

Notably, Ind-SO_4_ does appear in other high-performing combinations reported in Table 2, but always in conjunction with different couples, such as 5-AVA and serotonin, or carnosine and hypoxanthine. These observations indicate that optimal classification performance is not solely driven by the inclusion of metabolites with significant univariate associations. Rather, they underscore the necessity of empirically testing metabolite combinations, as predictive value in multivariate models cannot be reliably inferred from univariate significance alone (*p*-adj < 0.05 in the heatmaps).

Taking the first combination in Table 2: 5-AVA, carnosine, CA, and CDCA, the linear discriminant function (LD1) derived from this combination was:LD1 = −3.3279353 * *e*_5-AVA_−1.9457430 * *e*_Carnosine_ + 0.9974459 * *e*_CA_ + 0.2201517 * *e*_CDCA_

*e*_5-AVA_, *e*_Carnosine_, *e*_CA_, and *e*_CDCA_ represent the residuals for 5-AVA, carnosine, CA, and CDCA, respectively. A fixed threshold of 0.58 was applied across all LDA models. Samples with scores above 0.58 were classified as AD, while those with scores below or equal to 0.58 were classified as controls. This combination yielded a sensitivity of 0.74, a specificity of 0.79, and an overall accuracy of 0.76. However, it is important to note that the models reported in Table 2 were not validated using cross-validation and should therefore be considered exploratory. As such, they may be affected by overfitting, particularly given the moderate sample size.

To address this issue, we applied Leave-One-Out Cross-Validation (LOOCV), a robust method that tests the model’s ability to generalize to new, unseen data. While none of the four-metabolite combinations achieved an accuracy ≥0.75 under LOOCV, several retained high sensitivity (up to 0.78), as reported in Table 3. Notably, one such combination included carnosine, homocysteine (HCys), hypoxanthine, and Ind-SO_4_; another comprised 5-AVA, CA, serotonin, and Ind-SO_4_. Following LOOCV, the best-performing combinations also included metabolites that both tend to increase in AD and that tend to decrease. Notably, some combinations present in Table 2 were not retained after LOOCV (Table 3), indicating that validation filtered out overfitted models. However, four combinations were consistently effective across both evaluations, including (i) carnosine, phenylacetylglycine (PAG), TCA, and TMAO, (ii) 5-AVA, carnosine,1−Met−His, and CA, (iii) 5-AVA, CA, serotonin, and Ind-S04 and (iv) 5-AVA, carnosine, nitro-Tyr, and hypoxanthine.

#### 2.2.3. Multivariate Analysis: Five Metabolite Combinations

The same analytical pipeline was extended to evaluate combinations of five metabolites, resulting in over 18 million combinations tested (n = 18,474,840). As previously noted, adding a fifth metabolite to the top-performing four-metabolite models (Table 2) did not necessarily improve classification performance. Nonetheless, the results presented in Table 4 highlight the central role of 5-AVA and carnosine, which co-occur in 7 out of the 10 highest-scoring five-metabolite combinations. As observed in the four-metabolite models, all selected combinations included both metabolites that tend to increase in AD and those that tend to decrease in AD. The top-performing—comprising 5-AVA, carnosine, CA, hypoxanthine, and 3-IPA—include metabolites from four distinct biochemical families and achieved strong classification metrics, with sensitivity, specificity, and overall accuracy values all approaching 0.80. Notably, while carboxylic acids were absent from the top five-metabolite combinations, fatty acids, which were not present in any of the best-performing four-metabolite sets (Table 2), did appear in Table 4. In one instance, two of the five metabolites, eicosatrienoic acid (FA(20:3)) and eicosenoic acid (FA(20:1)), were fatty acids, with one showing a tendency to decrease and the other to increase in AD.

Model robustness was further evaluated using Leave-One-Out Cross-Validation (LOOCV), yielding notable results. Unlike the four-metabolite models, which did not reach accuracies ≥ 0.75 under LOOCV, several five-metabolite combinations maintained strong performance. In the case of combinations of five, cross-validation reinforced the potential of some of the combinations shown in Table 4. The combination of 5-AVA, carnosine, CA, hypoxanthine and 3-IPA, which was ranked first in Table 4 and belonged to four different families, appears fifth in Table 5, with notable specificity and sensitivity values of 0.79 and 0.73, respectively. The top-ranked combination following LOOCV was composed of 5-AVA, carnosine, c4−OH−Pro, CA, and FA(18:2). This model, which includes compounds from three different biochemical families, had previously ranked second in Table 4.

Taken together, these findings suggest that the most effective LDA-based classification models consistently incorporate residuals from amino acid-related compounds and bile acids. In particular, 5-AVA, carnosine, and CA emerged as highly promising features for discriminating between AD patients and controls, with five-metabolite combinations outperforming four-metabolite models following cross-validation.

## 3. Discussion

Diagnostic and prognostic tools are well-established for many diseases, including cancers, which often rely on specific tumor progression markers, and chronic conditions like diabetes, where plasma glucose and glycosylated hemoglobin serve as key indicators for diagnosis and monitoring [8]. In stark contrast, neurodegenerative diseases lack reliable diagnostic markers in body fluids and have no widely accepted measures for tracking disease progression [9].

Research efforts to improve diagnostics and prognostics in neurodegenerative diseases have primarily focused on protein-based biomarkers rather than small molecules. For AD, in particular, much attention has been directed toward aggregates of β-amyloid and phosphorylated tau as potential indicators.

Advances in metabolomics could transform this challenging landscape by offering two key advantages. Unlike traditional methods, metabolomics does not require a priori assumptions about disease mechanisms or pathways. The biomarkers identified may be so indirect that their biological relevance may only become clear over time, yet they could still significantly improve diagnostic and prognostic accuracy [10,11,12].

Equally crucial is the vast volume of data generated by metabolomics, which requires advanced computational tools and sophisticated analytical strategies to achieve reliable predictions in disease diagnosis and prognosis. While initial insights can often be drawn from univariate analyses, the inherent complexity and high dimensionality of metabolomic datasets typically demand multivariate approaches to enable robust and meaningful interpretation.

Multivariate analysis combined with a novel development based on concomitant parameters proved useful for interindividual comparisons using metabolomics from human tears, a body fluid with high inter-individual variability. The development devised an algorithm to predict whether a given tear sample came from a male or a female [6]. The same development using a more complex set of data from patients of AD and controls has led to the selection of metabolites that, when combined, may distinguish AD patients from controls with up to 75–80% accuracy. Diagnostic sensitivities of 0.75–0.80 would be particularly significant in neurodegenerative diseases. For example, in Parkinson’s disease, combinations of multiple diagnostic methods are required, yet they often achieve accuracies below 70% [13,14].

In Parkinsonian syndromes, dopamine transporter imaging using single-photon emission computed tomography (DaTscan) demonstrates high sensitivity (up to 0.85) and specificity (up to 0.9) in differentiating Parkinsonism from essential tremor. However, it lacks specificity in distinguishing Parkinson’s disease from atypical Parkinsonian disorders [15]. Similarly, while significant progress has been made in identifying biomarkers for AD, a definitive diagnosis still relies on postmortem examination of brain tissue to confirm characteristic neuropathological lesions.

Accuracies of 0.75–0.80 using plasma samples in which 6–8 metabolites are measured are also notable in comparison to procedures that are more interventional and/or that require expensive imaging techniques. Magnetic resonance imaging (MRI) is indeed instrumental in the diagnosis of multiple sclerosis, with specificity and sensitivity values that can be greater than 0.80 if McDonald criteria are applied, i.e., if dissemination in time and space is considered [16]. Results are more modest in the diagnosis of epilepsy using electroencephalogram with a 0.70–0.90 range for specificity and 0.70–0.85 for sensitivity [17].

In this report, some of the models identified through LOOCV demonstrated a trade-off between sensitivity and specificity. Although several combinations achieved high sensitivity (up to 0.78), their specificity was comparatively lower (as low as 0.65 in one instance), potentially leading to an increased rate of false positives. This limitation should be taken into account when aiming at clinical implementation, where high specificity is often crucial to avoid unnecessary follow-up procedures or misclassification. Our results highlight that adding the fifth parameter to a combination of four does not necessarily increase predictive accuracy. However, it is evident that a combination of five appropriately selected molecules has more predictive power than four. In conclusion, the more parameters considered, the more chances to obtain accuracies >0.9. Despite it being theoretically possible, it made little sense to increase the number of parameters to try to reach sensitivities and specificities greater than 0.90. Two factors were considered. One is that, considering economic and procedural terms, the number of parameters should be kept to a minimum to be useful and used for diagnosis/prognosis. In addition, with the dataset we used, which is based on relative values and not actual concentrations, it made little sense to make calculations with millions of combinations with 6, 7, or 8 metabolites.

Importantly, the metabolites most consistently identified across the top-performing classification models have been previously linked, either directly or via related biological pathways, to mechanisms potentially relevant to AD. This convergence supports the biological plausibility of our findings. For example, 5-aminovaleric acid (5-AVA), a lysine degradation product, has been implicated in gut–brain communication and microbiota-associated pathways, which are increasingly recognized as key players in neurodegeneration [18]. Carnosine, a dipeptide with well-established antioxidant and anti-aggregant properties, was found to be elevated in peripheral samples in our dataset, contrasting with previous reports of reduced central levels [19]. This discrepancy may reflect compensatory systemic mechanisms. Altered levels of cholic acid (CA), a primary bile acid involved in cholesterol metabolism and amyloid regulation, have also been associated with AD [20]. However, CA concentrations may be influenced by external factors such as medication use and microbiota composition. Hypoxanthine, a purine metabolism intermediate, has been associated with oxidative stress and cholinergic dysfunction [21]. Lastly, indoxyl sulfate (Ind-SO_4_), a gut-derived uremic toxin, is known to promote neuroinflammation and neuronal apoptosis through oxidative and inflammatory pathways, and has been studied in the context of cognitive impairment [22].

One of the limitations of our study is the lack of concentrations for the different metabolites. It is essential to have real concentration to make our procedure robust and propose algorithms with 4 to 6 (real) metabolite concentrations to check their performance in terms of sensitivity and specificity. In addition, selecting the most “powerful” algorithm would require validation using data from other cohorts.

While the diagnostic accuracy achieved by plasma metabolomics does not yet exceed that of established plasma protein biomarkers such as *p*-tau217 and GFAP [23], it is important to note that all reported accuracies are inherently relative to the diagnostic accuracy of the reference cohort. Since no current in vivo clinical classification achieves 100% certainty, performance comparisons between biomarkers must be interpreted with this limitation in mind.

Another limitation of our study—and of plasma metabolomics more broadly—is that current analytical platforms still do not reliably detect certain small molecule species, such as glycated or nitrated metabolites, which may ultimately prove more informative than many of the compounds currently measurable. In contrast, protein biomarkers like *p*-tau217 and GFAP [23] have benefited from over a decade of sustained technological refinement and clinical validation. The metabolomics field, however, remains in evolution. As detection technologies advance and molecular coverage expands—particularly to include underexplored metabolic derivatives—the diagnostic potential of metabolomics-based approaches may improve substantially, potentially matching or even surpassing that of established protein-based markers.

Future work should integrate actual concentration values to develop and rigorously validate algorithms based on four to six metabolite concentrations, assessing their sensitivity and specificity across diverse cohorts. Ultimately, such algorithms could prove transformative in discriminating between early-onset AD, mild cognitive impairment, prodromal AD, and full-blown AD, provided metabolites are carefully selected and determined.

The plasma levels of certain metabolites that are included in the *Biocrates MxP^®^* kit used in the Kalecký et al., (2022) study [5], particularly bile acids, are known to be highly sensitive to external factors such as diet, gut microbiota composition, and medication use. These variables were not controlled for in our study due to data unavailability, which may have contributed to the observed inter-individual variability and diminished the statistical significance of some findings. Future studies should aim to account for these confounding factors, whether through cohort selection, detailed metadata collection, or appropriate multivariate modeling. Any future validation efforts should be carried out using real (absolute) concentrations of metabolites. Validation on independent cohorts using only relative values may lead to inconsistent or non-reproducible results. Therefore, the next critical step is to apply our approach to datasets generated under similar platforms but reporting real concentrations, allowing us to refine and validate diagnostic algorithms based on quantifiable biomarkers across diverse populations.

## 4. Materials and Methods

### 4.1. Data Collection

This analysis is based on data from a previously published study [5], which employed a targeted metabolomics approach using the *Biocrates MxP^®^ Quant 500* system to quantify metabolites in plasma and cerebral cortex samples from AD patients and healthy individuals. For the present study, only plasma data were re-analyzed. The cohort included 94 AD patients and 64 control subjects, all aged ≥55 years, recruited in a longitudinal study conducted by the Texas Alzheimer’s Research and Care Consortium (TARCC) between 2005 and 2018 [24]. Participants were enrolled at dementia clinics within TARCC institutions; control subjects consisted of family members and volunteers. sj-csv-9-alz-10.3233_jad-215448.csv, and sj-csv-11-alz-10.3233_jad-215448.csv files, provided by Kalecký and cols. [5] as supplementary information, served as the basis for re-analysis. Two control samples (PX009 and PX025), originally included in these files, were excluded from the present analysis due to incomplete data. The datasets included demographic information (including sex, age, fasting time) and metabolite measurements obtained using the *Biocrates MxP^®^ Quant 500 system.* As detailed in the paper that reported the metabolomics data [5], AD diagnosis in the TARCC cohort was established following a clinical examination supported by a standardized neuropsychological battery. The diagnosis adhered to the NINCDS-ADRDA criteria, classifying participants as “probable AD” [25]. Only individuals with consistent diagnosis across at least three annual follow-up visits were included. Control subjects were cognitively intact, defined by a Clinical Dementia Rating (CDR) of 0 and absence of cognitive impairment. All assessments were conducted across TARCC consortium sites following harmonized procedures.

### 4.2. Data Handling

Metabolite quantification was performed using the *Biocrates MxP^®^ Quant 500* targeted metabolomics kit (Biocrates Life Sciences AG, Innsbruck, Austria). This platform quantifies up to 524 lipids using flow-injection analysis tandem mass spectrometry (FIA-MS/MS) and 106 small molecules through chromatography. Mass spectrometry data were acquired using a high-resolution liquid chromatograph coupled with a triple quadrupole tandem mass spectrometer (LC-MS/MS). Peak identification and area under the curve calculations were performed using the *Biocrates MetIDQ™ software* (Oxygen-DB110-3005 version). Additional details on data acquisition and generation procedures are detailed in the original Kalecký et al. study [5].

To enhance the comparability of metabolite concentrations across individuals, a novel predictive model was developed to account for demographic variability and improve data normalization. This model, previously used for the analysis of human tear metabolomics data [6], integrates non-clinical variables, age, sex and fasting time, along with a concomitant variable consisting of the concentration of a given metabolite, *metabolite_c*, which serves to predict the value of other metabolites within the same biochemical group. Logarithmic transformations were applied to stabilize variance and improve the model performance, while linear regression equations were constructed to estimate metabolite levels based on the selected *metabolite_c* and demographic data. As previously proved [6], this approach provided a robust framework for data normalization, overcoming the challenge of non-Gaussian distributions often encountered in metabolomics datasets of human body fluid samples.

It is important to note that data in [5] are not reported as concentrations in molar, which precludes direct absolute quantitative comparison. However, since the data are provided as area under the curve values and were generated using standardized acquisition and processing protocols, inter-individual comparisons remain valid within each metabolite.

### 4.3. Statistical Analysis

Statistical analyses were performed in *RStudio* (version 2024.09.0+375), using several packages including *dplyr* (v1.1.4) and *tidyr* (v1.3.1). Raw datasets underwent preprocessing to ensure the removal of low-quality data, i.e., entries with multiple missing values due to concentrations being “below the limit of detection (LOD)”. The initial dataset, obtained after removing samples PX009 and PX025, was further refined to include small molecule metabolites within eight biochemical groups: amino acids, amino acid-related (AA-related), bile acids, biogenic amines, indoles and derivatives, carboxylic acids, hormones and related compounds, and fatty acids (FAs). Outliers were identified within each biochemical group, based on metabolite levels exceeding three standard deviations from the mean. Samples exhibiting >10% outliers relative to the total metabolites in the group were excluded. This threshold was chosen to balance the removal of extreme values while preserving as much biological variability as possible.

### 4.4. Determination of the Concomitant Metabolite for Each Biochemical Family

An iterative approach was applied as previously described to determine the most suitable concomitant metabolite (*metabolite_c*) within each group [6].

The linear regression model was implemented in *R* using the following formula syntax:**log_10_(metabolite)_predicted_ ~ Age * Sex + Fasting Time + Group + log_10_(*metabolite_c*)**

In each iteration, one metabolite was chosen as *metabolite_c* and a nested loop iterated over all other metabolites within the same biochemical family. The final selection of the *metabolite_c* for each chemical family was based on a combination of statistical and biological criteria. Specifically, candidate metabolites were evaluated according to the overall explanatory power of the models in which they were included, as reflected by adjusted *R*^2^ values, as well as the significance of the full model, evaluated with F-test *p*-values. In addition to these statistical considerations, biological relevance also played a key role in the final selection.

### 4.5. Residual Correlation Heatmaps

The predicted values for each metabolite were obtained from the same regression model previously described, which includes age, sex, fasting time, group, and the group-specific concomitant metabolite as covariates. Residuals represent the portion of metabolite variability not explained by the covariates and were used to assess inter-metabolite relationships within each group. The Pearson correlation coefficient matrix of these residuals was then computed, excluding the concomitant metabolite itself. This approach aimed to capture co-variation patterns among metabolites that persist after accounting for known sources of variability, potentially indicating underlying biological interconnections. A correlation heatmap was generated to visualize the metabolite interdependence’s structure after covariates adjustment. Heatmaps were generated using the *corrplot* (v0.95), *ggplot2* (v3.5.2), and *reshape2* (v1.4.4) packages.

### 4.6. Linear Discriminant Analysis (LDA)

For Linear Discriminant Analysis (LDA), the model used in *R* to obtain residuals within each biochemical family was:**log_10_(metabolite)_predicted_ ~ Age * Sex + Fasting Time + log_10_(*metabolite_c*)**
i.e., the group (patient or control) was not considered. The residuals (***e***) calculated as:***e*_metabolite_ = log_10_(metabolite)_actual_ − log_10_(metabolite)_predicted_**

All were merged into a single data frame for classification, retaining only the common samples across all biochemical groups, and resulting in a final dataset of 139 samples (81 AD cases and 58 controls -NC-). To ensure that the classification was based solely on metabolic features, the residuals were used as predictors. LDA was then performed on these residuals to classify the samples into AD and control groups. Initially, each residual was tested individually, and subsequently, models were built using combinations of residuals—from two up to five metabolites—according to the *R* formula syntax:**Group ~** ***e*_1_ + *e*_2_** in the case of 2 metabolites (1 and 2) and, similarly,**Group ~** ***e*_1_ + *e*_2_ + *e*_3_ + *e*_4_ + *e*_5_** in the case of 5 metabolites (1 to 5).


Model performance was validated using Leave-One-Out Cross-Validation (LOOCV), where a sample was excluded from the training set and used as an independent test case. This procedure was repeated for all samples. The metrics computed to evaluate model performances included accuracy (proportion of correct predictions), sensitivity, and specificity.

As results of LDA were normalized and thresholds for categorization go from 0 to 1, different thresholds (i.e., probability cut-offs) were used to compute sensitivity, specificity, and overall accuracy. Thresholds of 0.1, 0.2, 0.3, 0.4, 0.5, 0.6, 0.7, 0.8, and 0.9 were applied to the predicted probabilities to determine class membership, aiming to achieve the best balance between sensitivity and specificity. Thresholds of 0.5 and 0.6 yielded better classification performance compared to the others. When values between 0.5 and 0.6 were tested, the optimal threshold was 0.58. This value aligns with the proportion of AD cases in the dataset, a behavior commonly observed in probabilistic classifiers, where the decision boundary often aligns with the underlying class distribution in class imbalance circumstances.

LDA-related computations were performed using the *MASS* (v7.3.65) and *caret* (v7.0.1) packages in R.

## Figures and Tables

**Figure 1 ijms-26-06991-f001:**
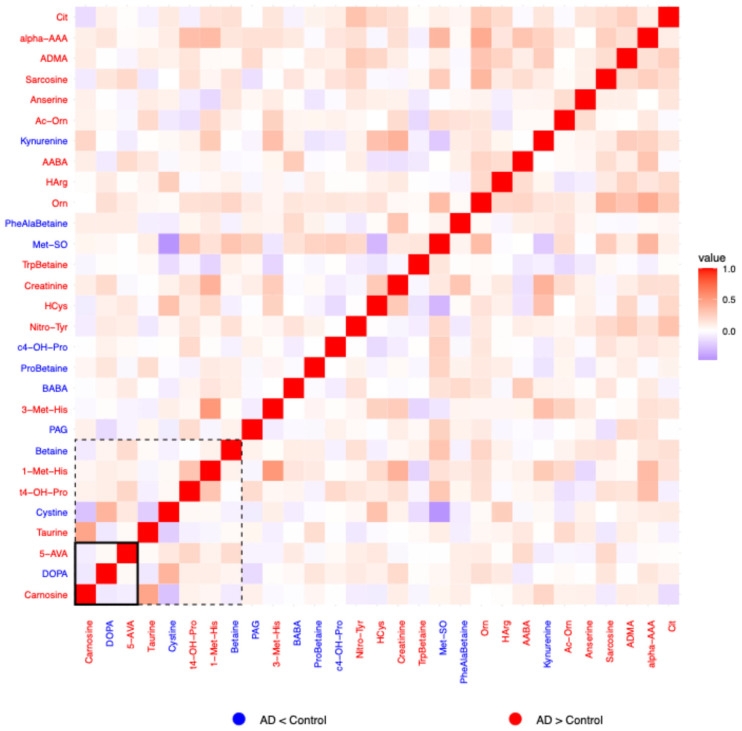
Heatmap of correlation of residuals of amino acid-related molecules on comparing data from patients and controls. *Metabolite_c* for this family was symmetric dimethylarginine (SDMA). Blue labels indicate tendency to decrease in AD and red labels indicate a tendency to increase in AD. Red cells indicate positive correlation (1 means perfect correlation), blue cells indicate negative correlation, and white cells indicate no correlation. *p* < 0.05 for residuals of metabolites within the dashed line frame. *P*_adjusted_ < 0.05 for residuals of metabolites within the solid line frame.

**Figure 2 ijms-26-06991-f002:**
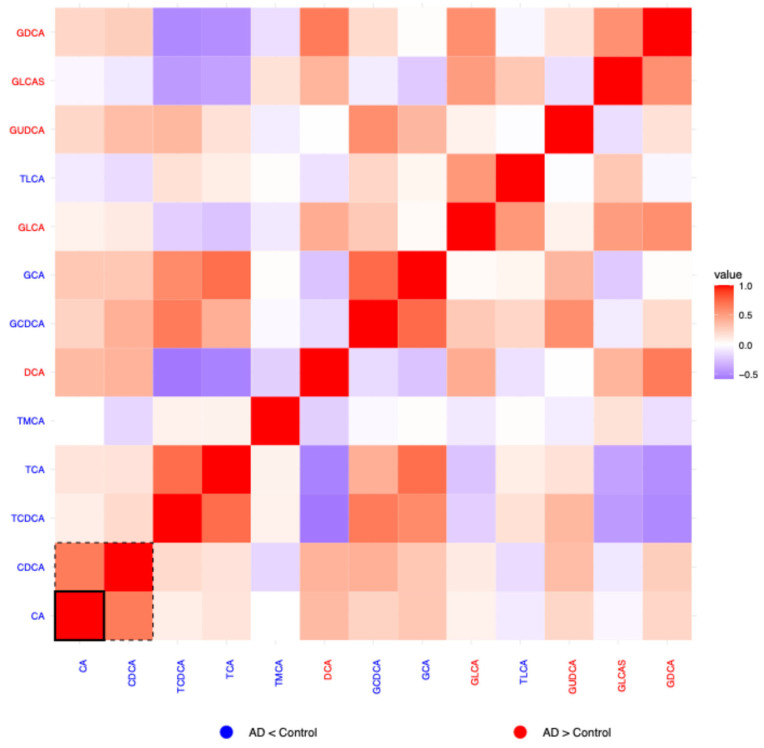
Heatmap of correlation of residuals of bile acids on comparing data from patients and controls. *Metabolite_c* for this family was taurodeoxycholic acid (TDCA). Blue labels indicate a tendency to decrease in AD and red labels indicate tendency to increase in AD. Red cells indicate positive correlation (1 means perfect correlation), blue cells indicate negative correlation, and white cells indicate no correlation. *p* < 0.05 for residuals of metabolites within the dashed line frame. *P*_adjusted_ < 0.05 for residuals of metabolites within the solid line frame.

**Figure 3 ijms-26-06991-f003:**
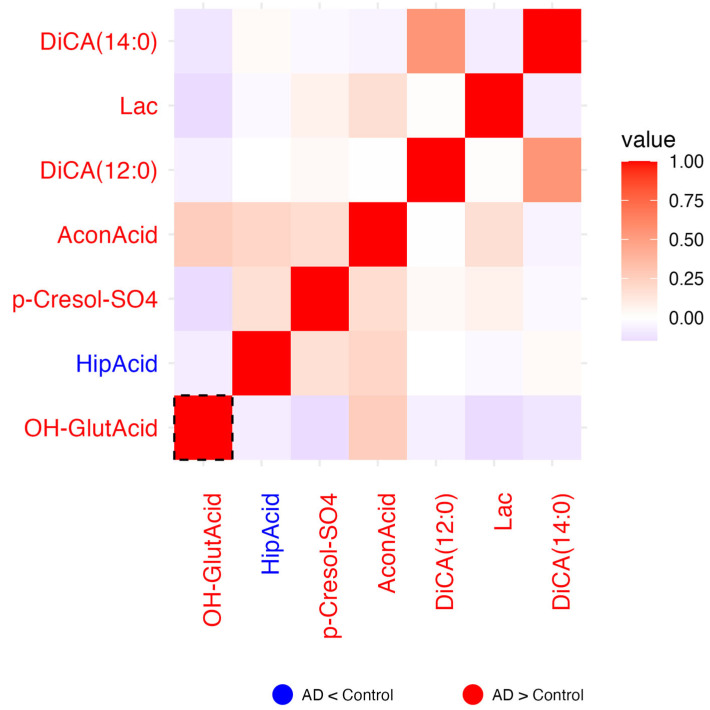
Heatmap of the correlation of residuals of carboxylic acids on comparing data from patients and controls. *Metabolite_c* for this family was Succinic Acid (Suc). Blue labels indicate tendency to decrease in AD and red labels indicate a tendency to increase in AD. Red cells indicate positive correlation (1 means perfect correlation), blue cells indicate negative correlation, and white cells indicate no correlation. *p* < 0.05 for residuals of metabolites within the dashed line frame.

**Figure 4 ijms-26-06991-f004:**
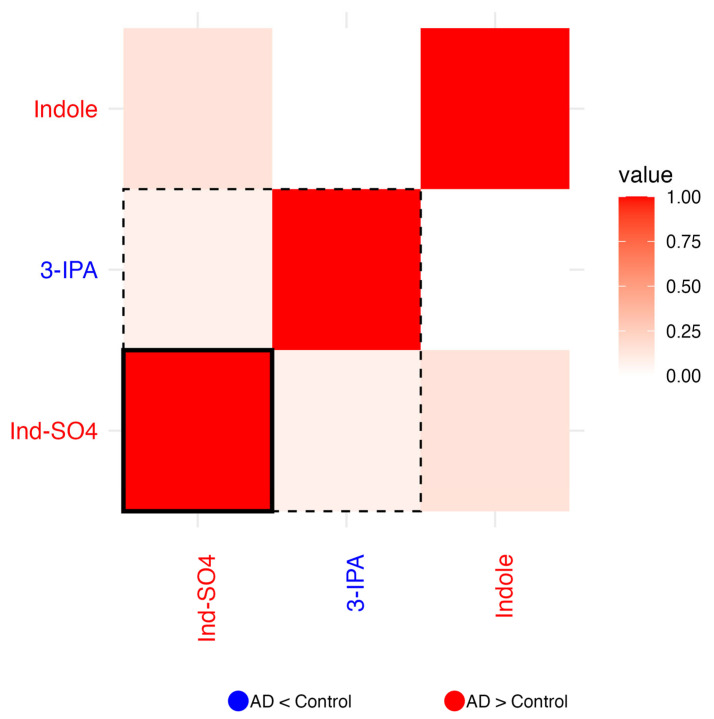
Heatmap of correlation of residuals of indoles on comparing data from patients and controls. *Metabolite_c* for this family was 3-Indoleacetic acid (3-IAA). Blue labels indicate tendency to decrease in AD and red labels indicate a tendency to increase in AD. Red cells indicate positive correlation (1 means perfect correlation), blue cells indicate negative correlation, and white cells indicate no correlation. *p* < 0.05 for residuals of metabolites within the dashed line frame. *P*_adjusted_ < 0.05 for residuals of metabolites within the solid line frame.

**Table 1 ijms-26-06991-t001:** Table reporting the statistical metrics (average) associated to the models with the selected concomitant within each biochemical group.

Group	Model ^a^	R^2^	F Test-*p*-Value
AA-related	log_10_(metabolite)_predicted_ ~ Group + Age * Sex + Fasting_time + log_10_(SDMA)	0.121	0.099
Bile Acids	log_10_(metabolite)_predicted_ ~ Group + Age * Sex + Fasting_time + log_10_(TDCA)	0.309	0.013
Biogenic Amines	log_10_(metabolite)_predicted_ ~ Group + Age * Sex + Fasting_time + log_10_(Spermine)	0.112	0.101
Carboxylic Acids	log_10_(metabolite)_predicted_ ~ Group + Age * Sex + Fasting_time + log_10_(Succinic)	0.108	0.134
Indoles	log_10_(metabolite)_predicted_ ~ Group + Age * Sex + Fasting_time + log_10_(3-IAA)	0.054	0.188
Fatty Acids	log_10_(metabolite)_predicted_ ~ Group + Age * Sex + Fasting_time + log_10_(DHA)	0.294	0.014

^a^ The selected concomitant metabolite (*metabolite_c*) is in red.

**Table 2 ijms-26-06991-t002:** Top 10 combinations of four metabolites that yield classification accuracies of 0.75 or higher without Leave-One-Out Cross-Validation. The table includes the metabolic families of each metabolite and their trend in AD (blue decrease, red increase).

AA-Related	Bile Acids	Biogenic Amines	Carboxylic Acids	Indoles	Fatty Acids	Accuracy	Sensitivity	Specificity
5-AVA, Carnosine	CA, CDCA					0.76	0.74	0.79
5-AVA, Carnosine, c4−OH−Pro	CA					0.76	0.78	0.74
5-AVA, Carnosine,1−Met−His	CA					0.76	0.76	0.76
5-AVA	CA	Serotonin		Ind-SO_4_		0.75	0.8	0.69
Carnosine, Orn	CA	Hypoxanthine				0.75	0.73	0.79
5-AVA, Carnosine	CA		Lactic			0.75	0.74	0.77
Carnosine, PAG	TCA	TMAO				0.75	0.74	0.77
5-AVA, Carnosine, Nitro-Tyr		Hypoxanthine				0.75	0.74	0.77
Carnosine	CA	Hypoxanthine		Ind-SO_4_		0.75	0.78	0.72
5-AVA, Carnosine	CA		Hippuric			0.75	0.78	0.72

**Table 3 ijms-26-06991-t003:** Top 10 combinations of four metabolites that yield notable classification accuracies using Leave-One-Out Cross-Validation. The table includes the metabolic families of each metabolite and their trend in AD (blue decrease, red increase).

AA-Related	Bile Acids	Biogenic Amines	Carboxylic Acids	Indoles	Fatty Acids	Accuracy	Sensitivity	Specificity
Carnosine, Nitro-Tyr		Hypoxanthine		Ind-SO_4_		0.74	0.76	0.71
5-AVA, Carnosine	CA		Lac			0.74	0.74	0.74
Carnosine, HCys		Hypoxanthine		Ind-SO_4_		0.73	0.78	0.67
5-AVA, Carnosine	CA, TMCA					0.73	0.73	0.72
5-AVA, Carnosine, 3−Met−His	CA					0.73	0.73	0.72
5-AVA	CA	Serotonin		Ind-SO_4_		0.73	0.78	0.65
5-AVA, Carnosine	CA, GUDCA					0.73	0.75	0.69
5-AVA, Carnosine, Nitro-Tyr		Hypoxanthine				0.73	0.72	0.74
Carnosine, PAG	TCA	TMAO				0.73	0.73	0.72
5-AVA, Carnosine,1−Met−His	CA					0.73	0.75	0.69

**Table 4 ijms-26-06991-t004:** Top 10 combinations of five metabolites that yield classification accuracies of 0.78 or higher without Leave-One-Out Cross-Validation. The table includes the metabolic families of each metabolite and their trend in AD (blue decrease, red increase).

AA-Related	Bile Acids	Biogenic Amines	Carboxylic Acids	Indoles	Fatty Acids	Accuracy	Sensitivity	Specificity
5-AVA, Carnosine	CA	Hypoxanthine		3-IPA		0.78	0.78	0.79
5-AVA, Carnosine, c4−OH−Pro	CA				FA(18:2)	0.78	0.8	0.76
5-AVA, Carnosine	CA				FA(20:3), FA(20:1)	0.78	0.76	0.81
Carnosine, Nitro-Tyr, Harg		Hypoxanthine		Ind-SO_4_		0.78	0.81	0.72
5-AVA, Carnosine, c4−OH−Pro	CA				FA(18:1)	0.78	0.8	0.76
Carnosine, Orn, ADMA	CA	Hypoxanthine				0.78	0.75	0.81
5-AVA, Carnosine, c4−OH−Pro,PAG	CA					0.78	0.79	0.76
Carnosine, Nitro-Tyr, Cit		Hypoxanthine		Ind-SO_4_		0.78	0.79	0.76
5-AVA, Carnosine, Nitro-Tyr, ProBetaine	CA					0.78	0.78	0.77
5-AVA, 3−Met−His, t4−OH−Pro	CA	Xanthine				0.78	0.79	0.76

**Table 5 ijms-26-06991-t005:** Top 10 combinations of five metabolites that yield classification accuracies of 0.75 or higher using Leave-One-Out Cross-Validation. The table includes the metabolic families of each metabolite and their trend in AD (blue decrease, red increase).

AA-Related	Bile Acids	Biogenic Amines	Carboxylic Acids	Indoles	Fatty Acids	Accuracy	Sensitivity	Specificity
5-AVA, Carnosine, c4−OH−Pro	CA				FA(18:2)	0.76	0.78	0.74
5-AVA, Met−SO	CA	Serotonin		Ind-SO_4_		0.76	0.79	0.71
Carnosine, Orn, alpha-AAA	CA	Hypoxanthine				0.76	0.75	0.76
Carnosine	CA	Hypoxanthine	DiCA(14:0)	Ind-SO_4_		0.76	0.79	0.71
5-AVA, Carnosine	CA	Hypoxanthine		3-IPA		0.76	0.73	0.79
Carnosine, Nitro-Tyr		Hypoxanthine		Ind-SO_4_	FA(20:3)	0.76	0.79	0.71
Carnosine, Orn, PAG	CA	Hypoxanthine				0.76	0.75	0.76
5-AVA, Carnosine, c4−OH−Pro	CA		DiCA(14:0)			0.75	0.78	0.71
5-AVA, Carnosine, PAG	CA		OH−GlutAcid			0.75	0.75	0.74
Carnosine, Nitro-Tyr, c4−OH−Pro		Hypoxanthine		Ind-SO_4_		0.75	0.78	0.71

## Data Availability

All data used in the analysis is available in Appendix A.

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
