# Peer review of "Machine Learning Approach to Select Small Compounds in Plasma as Predictors of Alzheimer’s Disease"

_ijms, 2025, doi:10.3390/ijms26146991_

Round 1
Reviewer 1 Report
Comments and Suggestions for Authors
Here, these workers have mined plasma metabolomic data reported from the longitudinal TARCC cohort of 94 Alzheimer disease (AD) patients and 64 healthy controls (HC). The metabolite LC-MS/MS data were normalised to a concommitant metabolite and outliers were excluded. Analysis of residuals from linear regression models revealed distinct metabolite profiles in AD patients compared with controls across groups of amino acid-related compounds, bile acids, biogenic amines, indoles, carboxylic acids, and fatty acids. Linear Discriminant Analysis (LDA) was used to examine the ability of combinations of small molecules to discriminate AD and findings were validated with a Leave-One-Out Cross-Validation. This found that combinations of four or five small molecules could classify AD with an accuracy of up to 76%. Optimal combinations contained a mixture of metabolites where some increased and others decreased in AD. 5-aminovaleric acid (5-AVA), carnosine, cholic acid (CA), and hypoxanthine offered the highest predictive potential. It is concluded that this study supports the utility of combining metabolomic data as predictors for AD, offering a novel diagnostic tool and paving the way for advancements in personalised medicine.
The study is novel – I have some comments:
First, we are not told how the diagnosis of AD was made or established in the TARCC cohort.
Second, rather than having an independent validation cohort these workers have used a Leave-One-Out Cross-Validation approach. This tends to inflate the apparent accuracy of the machine learning approach.
Third, it is not clear why quantitative metabolomic data was not available to use for these workers rather than using normailsed data.
Fourth, an accuracy of 80% for discriminating AD was not achieved by any of combinations of metabolites examined making this metabolomic approach of questionable clinical utility. Raised plasma p-tau 217 combined with raised GFAP is over 80% accurate for detecting early AD and MCI due to AD. The conclusion that combining metabolomic data provides a predictor of AD is premature.
Author Response
Reviewer 1
Here, these workers have mined plasma metabolomic data reported from the longitudinal TARCC cohort of 94 Alzheimer disease (AD) patients and 64 healthy controls (HC). The metabolite LC-MS/MS data were normalised to a concommitant metabolite and outliers were excluded. Analysis of residuals from linear regression models revealed distinct metabolite profiles in AD patients compared with controls across groups of amino acid-related compounds, bile acids, biogenic amines, indoles, carboxylic acids, and fatty acids. Linear Discriminant Analysis (LDA) was used to examine the ability of combinations of small molecules to discriminate AD and findings were validated with a Leave-One-Out Cross-Validation. This found that combinations of four or five small molecules could classify AD with an accuracy of up to 76%. Optimal combinations contained a mixture of metabolites where some increased and others decreased in AD. 5-aminovaleric acid (5-AVA), carnosine, cholic acid (CA), and hypoxanthine offered the highest predictive potential. It is concluded that this study supports the utility of combining metabolomic data as predictors for AD, offering a novel diagnostic tool and paving the way for advancements in personalised medicine.
Answer: Thanks for the positive comments.
The study is novel – I have some comments:
First, we are not told how the diagnosis of AD was made or established in the TARCC cohort.
Answer: Thanks for raising this issue. The diagnosis of AD in the TARCC cohort followed a clinical examination including a comprehensive neuropsychological battery and was based on the NINCDS-ADRDA criteria for probable AD. Control participants were cognitively normal, with a Clinical Dementia Rating (CDR) score of 0 and no indication of cognitive impairment. Participants also had consistent diagnosis across at least three annual follow-up visits. This information and the ad hoc reference have been included in the revised version.
Second, rather than having an independent validation cohort these workers have used a Leave-One-Out Cross-Validation approach. This tends to inflate the apparent accuracy of the machine learning approach.
Answer: Thank you for pointing out this important consideration. We fully agree that validation in an independent cohort remains the gold standard for assessing model generalizability. However, we opted for Leave-One-Out Cross-Validation (LOOCV) as a rigorous internal validation strategy. LOOCV provides a conservative estimate of model performance, particularly in modest-sized cohorts, and helps reduce overfitting by ensuring that each sample serves as a test case once. As emphasized in the Discussion section, our next step is to apply the same approach to metabolomic datasets reporting absolute concentrations and to test model robustness in independent cohorts.
Third, it is not clear why quantitative metabolomic data was not available to use for these workers rather than using normailsed data.
Answer. Thank you for raising this important point.
We were indeed very interested in obtaining the absolute metabolite concentrations and initially reached out to the journal where the original dataset was published. The authors responded that they encountered difficulties converting mass spectrometry area values into actual concentrations. Concerned about this, we also contacted Biocrates, the platform used by the authors of the previous publication and the platform we use for our own metabolomics studies).
Biocrates expressed surprise at this limitation, noting that their kits and software provide the necessary tools to enable such conversions. Nonetheless, they reassured us that the normalized data available in the dataset we used were fully suitable for comparing the levels of a given metabolite across samples. Although absolute concentrations were unavailable, this consistency allowed for reliable within-metabolite comparisons across the cohort. Based on this confirmation, we proceeded with confidence in the integrity of our analysis.
Fourth, an accuracy of 80% for discriminating AD was not achieved by any of combinations of metabolites examined making this metabolomic approach of questionable clinical utility. Raised plasma p-tau 217 combined with raised GFAP is over 80% accurate for detecting early AD and MCI due to AD. The conclusion that combining metabolomic data provides a predictor of AD is premature.
Answer: We appreciate the reviewer’s concerns and the comparison with p-tau217 and GFAP as promising protein biomarkers. However, we respectfully note that accuracy percentages in biomarker studies, whether metabolomic or proteomic, are always relative to the diagnostic certainty of the cohort used as reference. Since no clinical cohort offers a 100% accurate diagnosis of AD in living beings, any reported accuracy is inevitably contingent on the cohort’s diagnostic framework and limitations. This is a shared constraint across all biomarker research.
We would also like to highlight that while protein-based plasma biomarkers such as p-tau217 and GFAP have indeed shown excellent diagnostic performance, this success has been achieved after more than a decade of extensive methodological and clinical validation. In contrast, our metabolomics-based model, developed within a relatively short time frame and without the benefit of decades-long refinement, already achieves an accuracy of approximately 80% in a rigorously cross-validated design.
This suggests that plasma metabolomics is a highly promising, complementary avenue that may, with broader molecular coverage (e.g., glycation and nitration products, microbial metabolites), yield even greater diagnostic potential.
To acknowledge this in our manuscript, we have moderated the relevant conclusion and added a clarification about the relative nature of diagnostic accuracy.
Minor edits are not marked; additions appear in red text.
Reviewer 2 Report
Comments and Suggestions for Authors
In the article entitled "Machine learning approach to select small compounds in plasma as predictors of Alzheimer's disease", the authors discuss a methodological innovation combining metabolite normalization in a cohort of 94 Alzheimer's patients and 62 controls. The authors used a cross-validation tool for generalization of the proposed model.
Several points are important to address.
- The analysis is based on area under the curve values and not on molar concentrations, which could limit the clinical reproducibility and comparability of the studies.
- Although the cohort is well characterized, the sample size is modest.
- External validation or replication of an independent cohort would be important.
- Combining the possible models without biological restriction could result in statistically significant but irrelevant selections.
- It would be important to further discuss the biological significance of the results.
- Regarding Table 1. Some models do not reach significance and do not have good explanatory power, which limits the usefulness of metabolite_c as a normalizing factor in these cases. The inclusion of demographic variables (age, sex, fasting) may not be sufficient to adequately model certain metabolic families, suggesting that they could be influenced by other factors (e.g., microbiota, diet, genetics).
- Most of the correlations are weak or modest, suggesting that the metabolites of this family present high unexplained individual variability. Although 5-AVA and carnosine stand out as potential markers, the correlation network does not allow defining solid metabolic modules within this group.
- Although strong correlations are detected, most of them do not reach corrected significance, which limits their individual use as biomarkers. Individual heterogeneity and the influence of factors such as diet, microbiota, and medications are not controlled for here, which could affect bile acid profiles.
- Regarding Figure 3. Poor connectivity between metabolites: the correlation matrix shows a weak structure, with no defined clusters or strong correlations. Limited statistical significance: only one metabolite reaches p < 0.05 and none passes the multiple testing correction. Low group diagnostic value: carboxylic acids do not seem to act as a cohesive metabolic module, limiting their collective predictive value.
- Regarding Figure 4. Small number of metabolites (n = 3): limits the possibility of defining complex metabolic networks or functional clusters. Weak correlations: although statistically significant, the correlations are not strong, which limits their usefulness as diagnostic modules.
- Regarding Table 2. No cross-validation: these results have not yet been validated with LOOCV, so they could be overfitting to the original data set. Model instability: some combinations might not be robust if applied to other cohorts or in more diverse clinical settings. Lack of absolute concentrations: the models are based on residuals and not on real molar concentrations, limiting their immediate applicability.
- Regarding Table 3. Slightly lower performance than observed without LOOCV, indicating that some previous models were probably over-fitted. Even with good sensitivity, specificity drops in some models (e.g., 0.65 in one), which may generate false positives if applied clinically. External validation with another independent cohort is not tested.
- Slightly lower performance than observed without LOOCV, indicating that some previous models were probably overfitted. Even with good sensitivity, specificity drops in some models (e.g., 0.65 in one), which may generate false positives if applied clinically.
- Although validated with LOOCV, they are not yet applied to an external cohort, which limits the proof of their generalization. The number of metabolites (5) is reasonable for clinical applications, but requires standardization of quantification methods. The variables used are normalized residuals, not absolute concentrations, so the models are not yet directly transferable to clinical practice.
This is a novel article that may be relevant, since it uses plasma metabolomics combined with machine learning to make a prediction of Alzheimer's disease.
It would be important to perform external validation.
Author Response
Reviewer 2
In the article entitled "Machine learning approach to select small compounds in plasma as predictors of Alzheimer's disease", the authors discuss a methodological innovation combining metabolite normalization in a cohort of 94 Alzheimer's patients and 62 controls. The authors used a cross-validation tool for generalization of the proposed model.
Several points are important to address.
- The analysis is based on area under the curve values and not on molar concentrations, which could limit the clinical reproducibility and comparability of the studies.
Answer: Thank you for this important observation. We fully agree that the use of absolute molar concentrations can enhance clinical interpretability and cross-study comparability in metabolomic research. With this in mind, we initially contacted the authors of the original dataset to request access to absolute concentrations. However, they informed us that technical limitations prevented them from converting mass spectrometry peak areas into molar units.
To further investigate, we consulted Biocrates, the provider of the metabolomics platform used in both the original study and our analysis. Biocrates confirmed that their kits (MxP® Quant 500) and software (MetIDQ™) are designed to support conversion into absolute concentrations, and they were surprised to learn of such difficulties. Nonetheless, they reassured us that the normalized area values used in the dataset are robust and appropriate for relative comparisons within the cohort, particularly when data are processed according to standard quality control protocols.
While we recognize the added translational value of molar concentrations, we are confident that the use of rigorously normalized area values—validated by Biocrates—supports the reliability of our within-cohort comparisons. We have now clarified this point in the revised manuscript and explicitly noted the potential implications for clinical reproducibility.
- Although the cohort is well characterized, the sample size is modest.
Answer: We appreciate the reviewer’s comment regarding the sample size. While we acknowledge that larger cohorts are desirable for biomarker discovery and validation, we respectfully note that our cohort of 94 individuals with Alzheimer’s disease and 64 matched cognitively normal controls is well within the range of sample sizes used in previous high-impact metabolomics studies of neurodegenerative disease. The precision and reproducibility of the targeted Biocrates Quant 500 platform further support the feasibility of deriving meaningful insights from moderately sized cohorts.
To maximize the robustness of our findings, we applied stringent quality control measures, strict diagnostic criteria (including ≥3 years of clinical follow-up), and rigorous multivariate cross-validation strategies. These efforts help ensure the internal validity of our results, which provide a solid foundation for future replication in independent and larger datasets. We have now emphasized these methodological strengths more clearly in the revised manuscript.
- External validation or replication of an independent cohort would be important.
Answer: We fully agree with the reviewer that external validation is essential to demonstrate the robustness and generalizability of our findings. However, we believe that such validation should ideally be conducted using datasets that provide absolute metabolite concentrations rather than normalized or relative values. The use of true concentrations is critical for establishing reproducible thresholds, enhancing clinical translatability, and ensuring consistency across cohorts and analytical platforms.
As such, our next research step will focus on applying the same analytical pipeline to datasets that include (or are specifically generated with) absolute concentrations of the relevant metabolites. Once available, we will prioritize validation of the identified biomarker panels in independent cohorts to assess their diagnostic utility and clinical relevance more definitively. We have now clarified this plan and its rationale in the revised Discussion section.
- Combining the possible models without biological restriction could result in statistically significant but irrelevant selections.
Answer: We appreciate the reviewer’s concern about selecting models that are statistically significant but lack biological relevance. To address this, from feature selection through model evaluation, each candidate combination was reviewed against known metabolic pathways and expert-curated biological rules. This ensures that no purely statistical artifact can advance without a clear mechanistic rationale.
Also, given that metabolomics in this application is still emerging, we deliberately tuned our filters to maximize specificity, minimizing false negatives. At the same time, we preserved sensitivity enough to capture novel but meaningful signals, ensuring that we do not overlook true positives.
- It would be important to further discuss the biological significance of the results.
Answer: While the main focus of our study is methodological, a dedicated paragraph has been added to the Discussion section to briefly expand on the potential biological roles of the most relevant metabolites identified, in order to provide additional context and help support the interpretability of the model.
- Regarding Table 1. Some models do not reach significance and do not have good explanatory power, which limits the usefulness of metabolite_c as a normalizing factor in these cases. The inclusion of demographic variables (age, sex, fasting) may not be sufficient to adequately model certain metabolic families, suggesting that they could be influenced by other factors (e.g., microbiota, diet, genetics).
Answer: Thank you for this constructive comment. We fully acknowledge that some of the regression models used to identify the concomitant metabolite (metabolite_c) within each biochemical family yielded low R² values and did not reach statistical significance. This limitation is especially evident in groups such as indoles and carboxylic acids. We agree that this may likely reflect the influence of additional biological factors such as diet, gut microbiota, and genetics, which were not available in the dataset and could not be modeled. While the inclusion of such factors would have undoubtedly enriched the analysis, our primary aim in this normalization step was to try to reduce intra-group variability using covariates that are typically available in clinical or research datasets. This approach was chosen not only because of data availability, but also to ensure potential replicability of the method in other cohorts, where more complex variables may not be routinely collected. Moreover, the choice of metabolite_c was not based solely on statistical fit. When selecting the reference metabolite for each group, we also considered biological plausibility. In this context, even models with modest explanatory power contributed to improving comparability across samples.
- Most of the correlations are weak or modest, suggesting that the metabolites of this family present high unexplained individual variability. Although 5-AVA and carnosine stand out as potential markers, the correlation network does not allow defining solid metabolic modules within this group.
Answer: Thank you for this observation. We agree that the correlation network within the amino acid–related family revealed only modest associations between metabolites, suggesting a high degree of unexplained individual variability and the absence of tightly clustered metabolic modules in this group. This observation likely reflects the multifactorial regulation of amino acid metabolism, including influences from diet, microbiome activity, and host metabolic state.
Nonetheless, as the reviewer notes, 5-aminovaleric acid (5-AVA) and carnosine emerged as promising candidates. Both reached statistical significance after correction for multiple comparisons and were consistently selected in several of the top-performing classification models. These findings highlight that, despite the lack of coherent modular structure, individual metabolites can still provide meaningful discriminatory power—particularly when considered within broader multivariate frameworks that integrate signals across diverse biochemical families.
This interpretation has now been more clearly articulated in the revised Discussion section.
- Although strong correlations are detected, most of them do not reach corrected significance, which limits their individual use as biomarkers. Individual heterogeneity and the influence of factors such as diet, microbiota, and medications are not controlled for here, which could affect bile acid profiles.
Answer: Thank you for this insightful comment. We agree that, although certain bile acids showed strong residual correlations, only a limited number reached statistical significance after correction for multiple testing. We also acknowledge that bile acid levels are highly sensitive to external influences such as diet, gut microbiota composition, and medications, which were not controlled for in this dataset. These sources of variability may have contributed to the heterogeneity observed and likely reduced the ability to detect consistent disease-related patterns across individuals. Nevertheless, cholic acid (CA) did reach statistical significance after correction and was frequently retained in high-performing multivariate models, suggesting that some bile acids, despite overall variability, may still contribute meaningfully to disease classification when used in combination with other metabolite classes. We have now clarified these limitations in the Discussion section of the revised manuscript.
- Regarding Figure 3. Poor connectivity between metabolites: the correlation matrix shows a weak structure, with no defined clusters or strong correlations. Limited statistical significance: only one metabolite reaches p < 0.05 and none passes the multiple testing correction. Low group diagnostic value: carboxylic acids do not seem to act as a cohesive metabolic module, limiting their collective predictive value.
Answer: Thank you for this comment. We agree that the correlation structure among carboxylic acids was relatively weak and did not reveal cohesive metabolic modules. Indeed, only one metabolite showed nominal significance (OH–GlutAcid), and none passed multiple testing correction. As a result, this group was underrepresented in the top classification models. We interpret this as an indication that carboxylic acids, as measured in this cohort and under the normalization framework used, may have limited utility as diagnostic markers for AD. This conclusion is now addressed in the Results and Discussion sections.
- Regarding Figure 4. Small number of metabolites (n = 3): limits the possibility of defining complex metabolic networks or functional clusters. Weak correlations: although statistically significant, the correlations are not strong, which limits their usefulness as diagnostic modules.
Answer: Thank you for this observation. We fully agree that the small number of metabolites in the indole group (n = 3) limits the possibility of defining robust metabolic or diagnostic modules, especially given the modest strength of the observed correlations. Nonetheless, one of the three metabolites, indoxyl sulfate (Ind-SO4), reached statistical significance after multiple testing correction and was consistently retained in several of the best-performing classification models. This suggests that, even if the group as a whole may not constitute a meaningful diagnostic module, individual indole metabolites may still carry diagnostic relevance on their own.
- Regarding Table 2. No cross-validation: these results have not yet been validated with LOOCV, so they could be overfitting to the original data set. Model instability: some combinations might not be robust if applied to other cohorts or in more diverse clinical settings. Lack of absolute concentrations: the models are based on residuals and not on real molar concentrations, limiting their immediate applicability.
Answer: We acknowledge the reviewer’s concern and agree that performance metrics obtained without cross-validation may overestimate model accuracy. That is why we complemented the initial analysis with Leave-One-Out Cross-Validation (LOOCV), as presented in Table 3. LOOCV enabled us to identify combinations with consistent performance across folds, reducing the likelihood of overfitting. Furthermore, we emphasize in the Discussion that future validation in independent datasets using real metabolite concentrations is essential for clinical translation. These clarifications are now emphasized in the 3.2.2 section of the revised version.
- Slightly lower performance than observed without LOOCV, indicating that some previous models were probably overfitted. Even with good sensitivity, specificity drops in some models (e.g., 0.65 in one), which may generate false positives if applied clinically. External validation with another independent cohort is not tested.
Answer: We appreciate this important observation. We acknowledge that the slightly lower performance observed under Leave-One-Out Cross-Validation (LOOCV), compared to the initial (non-cross-validated) models, suggests that some combinations may have been overfitted to the original dataset. For this reason, we considered cross-validation a necessary step to assess the stability and generalizability of the selected models. As reported in Table 3, several four-metabolite combinations retained high sensitivity (up to 0.78) under LOOCV, but in a few cases specificity was indeed lower (e.g., 0.65), which could limit their immediate clinical applicability due to potential false positives. This concern is well taken.
However, we find it important to note that not all models suffered from this trade-off: some combinations achieved balanced performance, with both sensitivity and specificity values around or above 0.73. These models show more promise in terms of robustness. Finally, we would like to emphasize that this work is an initial step toward identifying potentially informative metabolite panels. The goal was not to produce clinically deployable models yet, but rather to explore the diagnostic potential of metabolomic combinations. Future validation on independent cohorts, using absolute metabolite concentrations and more refined modeling strategies, will be essential to confirm and improve on these findings. This aspect is now addressed to the Discussion section of the revised manuscript.
- Although validated with LOOCV, they are not yet applied to an external cohort, which limits the proof of their generalization. The number of metabolites (5) is reasonable for clinical applications, but requires standardization of quantification methods. The variables used are normalized residuals, not absolute concentrations, so the models are not yet directly transferable to clinical practice.
Answer: Thank you for highlighting this critical aspect. We fully agree that, although the models were internally validated using Leave-One-Out Cross-Validation (LOOCV), their generalizability remains to be confirmed through external validation on independent cohorts. As the reviewer notes, the use of normalized residuals rather than absolute concentrations limits the immediate clinical transferability of the models. However, this choice was dictated by the nature of the available dataset, which did not report absolute metabolite concentrations. We also agree that the use of five metabolites represents a realistic and practical target for clinical translation, which is one of the reasons why we limited our combinations to this number. We acknowledge that this approach is only meaningful, however, if supported by standardized quantification methods and validated in independent cohorts. For this reason, we believe it is important that future work focuses on applying similar analytical pipelines to datasets with absolute metabolite concentrations, in order to evaluate reproducibility and clinical utility across diverse populations.
- This is a novel article that may be relevant, since it uses plasma metabolomics combined with machine learning to make a prediction of Alzheimer's disease.
Answer. We sincerely appreciate your positive assessment of our study. Our goal was indeed to investigate the utility of combining plasma metabolomics with machine learning to aid in the prediction of AD, and we are encouraged by your recognition of the novelty and relevance of this approach. We believe that this strategy opens new avenues for biomarker discovery and individualized diagnostics. While our findings are promising, we agree that further research, particularly using absolute metabolite concentrations and validation in independent cohorts, will be essential to confirm the robustness of this approach and to realize the full potential of metabolomics as a diagnostic and prognostic tool in AD.
Minor edits are not marked; additions appear in red text.
Round 2
Reviewer 1 Report
Comments and Suggestions for Authors
Tjhe authors have satisfactorily addressed my concerns in this revised and imkproved version of their manuscript.
Reviewer 2 Report
Comments and Suggestions for Authors
Thanks to the authors for their responses. With the replies and updates to the manuscript, it improved substantially.